# Immigration Policy and State Power

**Yasha Daniel Maccanico** [1,2]

1    Statewatch, London EC4Y 1DH, UK; yasha@statewatch.org
2    Migration, Mobilities Bristol Specialist Research Institute, University of Bristol, Bristol BS8 1TU, UK

**Abstract:** An analysis of 20 years of official documents (1995–2014) and legislative acts at national and EU levels using Jessop's Strategic Relational Approach (SRA) offers insights into inherent structural flaws in the Justice and Home Affairs aspects of European and member state migration policies. Focusing on two triptychs (hierarchy, governance and government; state power(s), strategic selectivities and structures) and tracking their development clarifies that this policy field's purposes stray beyond migration management. In fact, the EU migration policy model was set up to be inherently expansive and is intimately linked to EU institutions and national governments striving to enhance their power(s). This is why apparent aberrations and unlawful acts by states amounting to a power grab have developed into an attack against normative frameworks including human rights. This article investigates whether European approaches to immigration policy at the EU and national levels currently pose a problem in terms of state power and authoritarianism due to inherently expansive tendencies and the serial production of problems and hierarchies. It offers a methodological, state-theoretical contribution to address a policy fix in which the EU and its member states appear caught, with harmful effects that spread beyond EU borders through externalisation.

**Keywords:** EU; immigration policy; policy studies; JHA; migration; human rights

## 1. Introduction

This article analyses the relationship between European immigration policy at the EU and national levels and authoritarian statism. It builds on two decades (1998–2021) of reporting and analysis for *Statewatch* by the author using official EU documents as empirical sources. The selected documentation covers the 1995–2014 period, although monitoring of the development under the 2015 European Agenda on Migration (EAM) shows that problematic trends identified using earlier documentation did not just continue, they worsened. This work draws an inherent link between Justice and Home Affairs (JHA hereafter) aspects of the migration policy field (activity to counter irregular migration) and authoritarianism. Its findings are reinforced by the implementation of the EAM that structurally subordinated positive values and superordinate legal instruments to strategic migration policy goals.

Policy steering documents and legislative acts from four five-year periods (Parts 1–4: 1995–1999; 2000–2004; 2005–2009; 2010–2014) at three levels make it possible to chart the genealogy of actions deployed to supposedly create a well-managed EU migration system. Throughout this period, EU institutions and bodies consistently sought to constitute migration as a *"problem"* [1] impossible to solve without far-reaching transformative action and investment in border security, enabling discrimination and penalising people whose rights were undermined. This policy field did not appear concerned with offering solutions. Evidence shows that ad hoc pragmatic migration management solutions used by member states (MSs) were discouraged for contradicting a model based on structural exclusion.

This article starts by questioning the motives and underlying premises of migration policy, suggesting that other factors may have a comparable impact on policy than its subject. The focus is on institutional workings at the national, supranational and intergovernmental levels and on sequential developments in policymaking, policy approaches and



policy implementation through legislation. Two chapters outline the conceptual framework, theoretical outlook and methodology used to tackle the research problem, drawing on Jessop's state-theoretical work and the Strategic-Relational Approach (SRA hereafter) to state power [2,3]. The research problem set out in Section 5 concerns the distance between this policy field's formal and substantive aspects. Research findings are outlined in sections on "hierarchy", "strategic selectivities" and "structures", drawn from two triptychs (see below) used to analyse selected official documents. The conclusion notes the value of using state power as an interpretative key to examine policy development over time in the context of an ambitious state-building exercise. The research findings detail inherently problematic features of immigration policy like in-built expansiveness and reluctance to accept limits to state power enshrined in national, EU and international normative frameworks. The scope and substance of successive acts undertaken to achieve this policy field's strategic objectives reveal the potential of an innovative methodology to investigate structural flaws and systemic imbalances. Analysing empirical sources produced over time at different levels using the SRA may advance state theory by arming it with substance to make sense of an immigration policy fix that may be more of a "problem" than those it should prevent.

## 2. Is EU Immigration Policy even about Migration?

The question that underlies this article is whether EU immigration policy is mainly about migration. This concern appears relevant due to its instrumental function to advance other motives (developing EU security structures, enacting a power grab to erode limits to state power, creating an EU identity) relevant to state theory, EU studies, security studies, sociology, law, politics, criminology and policy studies. The hypothesis that immigration was pre-determined and constituted as a problem to enable expansive social control and coercive state capabilities, allowing discrimination and discretional exercise of power against specified subjects, is borne by analysis of official policy documents and legislative acts. The migration policy field inherently entails the serial production of hierarchies to allow differential treatment for various population groups. By 2015, it was being used to structurally subordinate positive values such as human rights, the rule of law and the right to information to achieving strategic migration policy goals whose continuity emerges from the document selection (Tables 1–4, below). This sacrifice for the sake of *"effectiveness"* became explicit under the EAM, with the *"hotspot approach"* deployed in Greece and Italy in 2015 and a drive to *"restore the credibility of the EU return system"* by undermining legal safeguards and enhancing return capabilities. EAM implementation in Italy and Greece, alongside the externalisation of EU migration policy to third countries, was even worse for human rights and refugee protection than heralded in institutional accounts to justify and define an uncompromising action plan.

**Table 1.** Empirical documents, part 1 (1995–1999).

| | | |
|---|---|---|
| EUD1 | Council Regulation no. 2317/95 of 25 September 1995 (visa list) | 3 October 1995 |
| EUD2 | Council Recommendation of 22 December 1995 (combating illegal migration) | 10 January 1996 |
| EUD3 | Council Recommendation of 22 December 1995 (expulsions) | 10 January 1996 |
| EUD4 | Council Resolution of 26 June 1997 (unaccompanied minors) | 19 July 1997 |
| EUD5 | Council Resolution of 4 December 1997 (marriages of convenience) | 16 December 1997 |

**Table 1.** *Cont.*

| EUD6 | Presidency Conclusions-JHA—Cardiff | 15/16 June 1998 |
|---|---|---|
| NLIT1 | Turco-Napolitano law, Legge 6 marzo 1998, no. 40 | 12 March 1998 |
| PP1 | Tampere European Council Presidency Conclusions | 15/16 October 1999 |
| NLUK1 | Immigration and Asylum Act 1999 | 11 November 1999 |

**Table 2.** Empirical documents, part 2 (2000–2004).

| EUD7 | Council Regulation (EC) no. 2725/2000 of 11 December 2000 (Eurodac) | 15 December 2000 |
|---|---|---|
| EUD8 | EU/ACP Partnership Agreement, Cotonou, 23 June 2000 | 15 December 2000 |
| EUD9 | Council Directive 2001/51/EC of 28 June 2001 (carrier sanctions) | 10 July 2001 |
| EUD10 | Communication from the Commission to the Council and the European Parliament on a common policy on illegal migration | 15 November 2001 |
| NLIT2 | Bossi-Fini law, Legge 30 luglio 2002, no. 189 | 26 August 2002 |
| NLUK2 | Nationality, Immigration and Asylum Act 2002 | 7 November 2002 |
| EUD11 | Council Regulation (EC) no. 407/2002 of 28 February 2002 (Eurodac rules) | 5 March 2002 |
| EUD12 | Council Directive 2002/90/EC of 28 November 2002 (facilitation) | 5 December 2002 |
| EUD13 | Council Regulation (EC) no. 343/2003 of 18 February 2003 (asylum responsibility) | 25 February 2003 |
| EUD14 | Presidency conclusions, Brussels 12–13 December 2003, "III. Freedom, Security and Justice" | 12/13 December 2003 |
| EUD15 | Council Regulation (EC) no. 2007/2004 of 26 October 2004 (Frontex) | 25 November 2004 |

**Table 3.** Empirical documents, part 3 (2005–2009).

| PP2 | The Hague Programme: Strengthening Freedom, Security and Justice in the European Union | 3 March 2005 |
|---|---|---|
| EUD16 | Communications from the Commission. A Strategy on the External Dimension of the Area of Freedom, Security and Justice. | 12 October 2005 |
| EUD17 | Council Presidency Conclusions, Part IV and Annex I: Global Approach to Migration: Priority Actions Focusing on Africa and the Mediterranean | 15/16 December 2005 |
| EUD18 | Regulation (EC) No 562/2006 of the European Parliament and of the Council of 15 March 2006 (Schengen Borders Code) | 13 April 2006 |
| EUD19 | 2008 Council Conclusions annex-European Pact on Immigration and Asylum. | 24 September 2008 |
| EUD20 | Directive 2008/115/EC of the European Parliament and of the Council of 16 December 2008 (Returns Directive) | 24 December 2008 |
| NLUK3 | Borders, Citizenship and Immigration Act 2009, 21 July 2009 | 21 July 2009 |
| NLIT3 | Legge 15 luglio 2009, no. 94 (security package) | 24 July 2009 |

**Table 4.** Empirical materials, part 4 (2010–2014).

| | | |
|---|---|---|
| EUD21 | Regulation (EU) No. 265/2010 of the European Parliament and of the Council of 25 March 2010 (long stay visas) | 31 March 2010 |
| PP3 | The Stockholm Programme: An Open and Secure Europe Serving and Protecting Citizens | 4 May 2010 |
| EUD22 | Regulation (EU) No 439/2010 of the European Parliament and of the Council of 19 May 2010 (EASO) | 29 May 2010 |
| NLIT4 | Legge 2 Agosto 2011, no. 129 (reception of European norms) | 5 August 2011 |
| EUD23 | Communication from the Commission, The Global Approach to Migration and Mobility | 18 November 2011 |
| EUD24 | Regulation (EU) No. 1052/2013 of the European Parliament and of the Council of 22 October 2013, (Eurosur) | 6 November 2013 |
| EUD25 | Council Presidency Note to delegations on an effective EU returns policy | 26 February 2014 |
| EUD26 | Council conclusions on EU Return Policy, Justice and Home Affairs Council meeting, Luxembourg, 5 and 6 June 2014 | 6 June 2014 |
| NLUK4 | Immigration Act 2014. | 14 May 2014 |

Tables from Maccanico [4].

Increased funding, powers and personnel were assigned to Frontex (re-named European Border and Coast Guard, EBCG), the EU agency devoted to securing EU borders and influencing MS approaches and practices in an advisory role. The instrumental use of available data collected from MSs to justify intransigence and develop migration policy enforcement interventions became a structural feature when Frontex began operating in 2005. Frontex was deployed in a "crisis" setting in both countries as part of EU Regional Task Forces (in Catania and Piraeus) to cooperate with MS authorities and EU agencies in technical, monitoring and advisory roles that straddled the national and EU levels. Blunt practices of dubious legality proliferated, like pushbacks at sea and land borders, criminalisation of civil society actors who help migrants and an engineered search-and-rescue (SAR hereafter) void in the central Mediterranean [5,6].

This last feature made an intrinsic feature of migration policy over time explicit, that of unauthorised mobility from third countries towards the EU leading to death as an effect of policy and political choices. What Mbembe terms "necropolitics" [7] has long coexisted with "biopolitics" [8] in this field, yet previous strategies of denial and blame deflection [9,10] no longer apply. At this point, achieving strategic migration policy goals was expressly given primacy over other values to justify state crimes [11] against people on the move. Foucault's concept of "biopower" operationalises biopolitics, giving rise to structures and strategies to protect society by defining population groups for subjugation/control and administering limitations on people's lives. Necropolitics, at the extreme bounds of biopolitics, refers to various strategies to limit life and assert the right to kill. These include exposing people to death and forms of exclusion that introduce death among living persons by limiting their capabilities. Gramsci [12] identified two means of killing: The first is expressed by the verb "to kill"; the second way involves "making life impossible" for certain subjects and requires a host of accomplices in society, not just among state agents.

The right to life (art. 2 ECHR) and a duty to rescue inscribed in national penal law, international law of the sea and seafarers' own ethical code, were subordinated to goals like "making sea crossings more dangerous" and "undermining the traffickers' business model". To make things worse, a country (Libya) that lacks a "place of safety" (PoS) in which to disembark survivors (by definition in the 1974 SOLAS Convention on the safety

of life at sea a rescue must end in a PoS) due to notorious human rights abuses in migrant holding centres was assigned an overlarge SAR zone. In this way, the EU and MSs retreated from international waters to prevent people disembarking in the EU, where they may claim asylum. The EU and MSs disregarded their duties under international law and undermined the *non-refoulement* principle that they claim to uphold by assisting a proxy, the Italian- and EU-funded Libyan Coast Guard, to return people to torturous, inhuman and degrading conditions. European vessels are forbidden from returning people to Libya due to such conditions, but a strategy to circumvent the "problem" was devised [13]. The knowledge that downscaling the presence of EU naval assets (replaced by aerial surveillance) would lead to deaths has been shown using Frontex documents in a 2019 submission to the International Criminal Court in which EU and MS authorities are accused of "crimes against humanity" [14]. The allegations refer to deaths at sea through wilful omission of rescue, obstructing NGO rescue crews, delegating SAR duties to Libya and complicity with degrading and torturous detention practices.

Recognising the disruptive function of EU and national migration policy approaches and the gap between their claims and pursued objectives helps to explain incongruities in migration policy, law and practice. Analysing a heterogeneous selection of policy documents and legislative texts from the EU, intergovernmental and national levels relevant to preventing "irregular" migration over two decades using the SRA [2,3] provides clarity by relying on and advancing theory, backed up by substance. Beyond contested readings about the interaction between levels and notions of pluralism, these levels' shared features include an identifiable direction of travel: (a) expansive assertion of state power(s); (b) serial production of hierarchies; and (c) interventions and creation of structures to enforce (a) + (b).

A choice to focus on two triptychs was vindicated by the findings produced and their explanatory power at three levels: Supranational (EU); intergovernmental (government representations in the Council); and national (Italy and UK) in 1995–2014, before the June 2016 Brexit referendum. Two triptychs were used and coupled to structure this inquiry: Firstly, (1) hierarchy, (2) governance and (3) government; and secondly, (1a) state power(s), (2a) strategic selectivities and (3a) structures. Pairing them up to interrogate empirical material about expansiveness, continuity, state power(s) and authoritarianism provides evidence of how, and insights into why, immigration policy (rather than migration or migrants) causes systemic problems beyond its supposed scope. This structure was inspired by Jessop's work to resolve the conflict between "government" and "governance" -based approaches by embracing them both while adding a third element, hierarchy. This model follows Jessop's conceptualisation of the exercise of state power as "government and governance in the shadow of hierarchy" [3] through which the SRA draws on Poulantzas, Foucault and Gramsci to make sense of the EU as a "state project".

Evidence from after the document selection's timeframe conveys a sense of urgency about the speed, spread and extent that an in-built tendency to enable abuse and legal breaches identified in previous phases has taken on. Jessop's SRA accounts for this shift through notions of "multiscalar metagovernance" (applicable to the EAM and hotspot approach) and "depoliticalisation" (applicable to Frontex). The former involves an expansive role and growing influence of EU migration governance in response to supposed governance and government failures in MSs [2]. Depoliticalisation [15], the removal of policy goals from the political arena to develop technical and operative capabilities and modes of intervention, applies to Frontex as an agency (structure) single-mindedly dedicated to securing borders and migration management. Research has focused on Frontex's lack of executive powers, with less attention paid to its ideological function and in-built tendency to concoct new risks to advance its pursuit of strategic goals framed in absolute terms.

Deployment in Greece and Italy that was portrayed as solidarity to lend assistance to MSs facing a refugee crisis lowered standards by consolidating legal breaches and denial of rights. It appeared that two frontline MSs were disciplined for not complying with their fingerprinting duties regarding people who irregularly reached their territory, to enter their

details in the Eurodac database. This failure resulted from authorities knowing that the Dublin system makes countries of arrival responsible for people who enter EU territory through their borders. Any future asylum applications lodged would lead to returns, even if people had moved to other MSs. The priorities of this EU deployment included ensuring a "100% fingerprinting rate" and preventing "secondary movements". Not activating an existing emergency relocation mechanism to deal with large-scale arrivals allowed EU authorities to create a "pressure cooker" effect [16] to implement exceptional measures in an emergency maintained by trapping new arrivals in frontline MSs. A corollary of this strategy was to undermine constitutional protections and asylum adjudication procedures.

Bigo [17] views the "security-immigration continuum" as leading to supranational and transnational governance regimes in which "security professionals" play key roles, liable to disempower governments by challenging their performance and methods. The "globalisation of (in)security" ushered in a "transnational regime of truth" derived from links between relevant professions (politicians, judges, police, military, intelligence) that concerns state theory. Multiscalar metagovernance views crises as not necessarily being a problem if they contribute to developing EU structures and introducing emergency measures. Crises may be artificially created or maintained over time to pursue ulterior (structural, economic, ideological) motives. Jessop used the SRA to analyse systemic imbalances in emergency situations in the financial (US and Eurozone debt crises) and security fields, noting the subordination of positive interests to strategic goals driven by governance structures [15]. Scipioni [18] outlines a "failing forward" argument drawn from neofunctionalist and liberal intergovernmental approaches to suggest that incomplete, flawed agreements liable to produce crises are "integral to a cyclical process of EU integration", as exemplified by the 2015 refugee "crisis". Scipioni notes that policy fields like asylum and migration are locked into a "path of incremental changes".

After outlining the research problem's substance, the adopted investigative strategy and conceptual framework, and justifying the use of an SRA outlook applied to empirical documents, we now turn to state theory.

## 3. State Theory

The SRA draws on Poulantzas, Gramsci and Foucault concerning states and the exercise of power, notably authoritarian statism [19], hegemony, territorial scales and strategic conjunctures [12], pervasive power relations, biopolitics and the use of racism to enhance the sovereign's power [8]. The thread binding these sources together despite other points on which they diverge is their conceptualisation of power as relational (between states; between states and individuals; between the EU and MSs—and vice-versa; between states and economic power brokers; between people across society). Jessop builds on these readings to account for past, present and future developments in statehood, including the EU. The EU experiment in state-building through a supranational governance model is in a bind in a field (JHA policy) dominated by MSs (interior ministries and law enforcement) and an ambition to develop EU structures and methods (databases, agencies, regulations, normalised practices with structural implications). These two influences' interaction produces friction, but there is consensus about enhancing coercive state power(s) at different levels for various purposes in partnership with industry [20–23]. The interplay between levels and the instrumental use of statehood and "EU-hood" to advance agendas, shift blame and legitimate breaching rules and international law would suffice as a reason for state theory to return to the centre stage in academic research.

Bourdieu's account of the state at the *College de France* [24] provides a rich backdrop, including recognition of inherently fictional aspects that must be upheld to sustain the credibility needed to enable rule by consent rather than by imposition or tyranny. The "*Raison d'État*" (reason of state) concept allows states to act beyond formal limits to their powers for urgent security and public order reasons in JHA policy, into which migration and asylum policies were hard-wired in Part 1 (1995–1999; PP1, the 1999 Tampere Council conclusions). Measures adopted during an antiterrorist emergency phase after the 11 September 2001

attacks in the USA in Part 2 (2000–2004) consolidated this shift [17]. Bourdieu's account examines the roles played by the state's actual and symbolic power, noting the difficulty of conceptualising the state with precision. Despite the inherent possibility for states to act beyond their supposed limits in defence of high state interests, Bourdieu argues that such breaches require concurrent proclamations of the validity of rules and laws that are thus violated. The credibility of state views and acts is crucial, because its legitimacy arises from a quasi-religious fiction according to which it serves the common good. In fact, "*state acts*" (like reports by state-appointed commissions) should represent "*a point of view that is not a point of view, but rather, an ideal condensation of all points of view*". Bourdieu views administrative science as "*the discourse state agents produce about the state... a veritable ideology of public service and the common good*" [24][1]. It is hard for such aspects, crucial for the state's operative capabilities, to hold firm in the presence of evidently false claims and harmful outcomes for its claimed purposes that are maintained over time.

Claims that the EU respects and promotes human rights and international law are belied by its strategy and interventions to implement and enforce a restrictive migration policy model. Documents in Parts 3 and 4 (2005–2014) are spattered with evidence that credibility, rationality and compliance with EU foundational values ceased to be priorities. This makes it hard to trust the inherent quality of state acts, resulting in dissenting civil society actors seeking to mitigate and draw attention to harms caused by migration policy enforcement [25]. Governance and management strategies were used to promote and achieve overambitious goals driven by an uncompromising, inherently ideological, outlook. This policy field's unbalanced narrative and instrumental rationality were written into the 2004 Regulation to establish the Frontex agency for the control of external borders (EUD15, Part 2) to lend technical and operational support to MS activities against "irregularised"[2] migration. Frontex was tasked with promoting its role through analytical work in pursuit of absolute goals, in a context where human rights gradually became "problems", "pull factors" or "instrumental obstacles to return".

The SRA's sophisticated reading of state power(s) and the formal and substantive aspects of the state leads to a six-dimensional model that enriches a basic three-dimensional understanding of the state comprising a "state apparatus", a "population" and a "territory". Relevant sources include Gramsci's understanding of the state as "political society + civil society, that is, hegemony armoured by coercion" [12] (see note 1). Jessop considers Gramsci's role as a spatial theorist relevant due to the attention he pays to "place, space and scale" and the role of international forces and geopolitics in "overdetermining" a country's "internal balance of forces" [26]. Poulantzas' focus on the "materiality of the state" identified features pointing to "authoritarian statism", including "intensified state control . . . decline of the institutions of political democracy . . . and . . . curtailment of so-called 'formal' liberties" [19], updated by Jessop to account for contemporary developments [3]. Power shifts from legislative to executive branches and the narrowing of available options in the policy-making process were part of this updated picture. Foucault argued that analysis should address "the genealogy of relations of force, strategic developments and tactics", using "war and battle" rather than "language and signs" as references, and replacing "relations of meaning" with "relations of power" [27].

These sources' points of convergence provide the SRA's theoretical pillars: The state as a site of strategy; power as relational; and attention paid to systems of domination and scope for resistance [3]. Documents from the national (NL), supranational (EUD, PP) and intergovernmental (EUD, PP) levels pertaining to governance, government and hierarchy (MS dominance in the JHA field, distinction between population groups) shed light on these aspects and their interplay. State power(s), strategic selectivities and structures can be identified and tracked over time using official documents from decision-making, legislative and executive bodies at three levels. Structures are not just indicative of the state's institutional materiality, they show what proposals materialise and are built upon, whose workings entail consequences and effects.

Although structures mainly featured in NL documents in Part 1 (1995–1999), from Part 2 to Part 4 (2000–2014), structures were also developed at the EU level (Eurodac, EUD7, 11, 13; Frontex, EUD15; EASO, EUD22; Eurosur, EUD24) for migration policy enforcement purposes. Strategic selectivities in the JHA policy field emerge clearly from three mid-term programmes to develop the Schengen Area of Freedom, Security and Justice (AFSJ, Tampere, 1999, PP1; The Hague, 2005, PP2; and Stockholm, 2010, PP3) and other policy-steering documents. The continuity on show suggests that strategic migration policy goals and their expansive vocation barely shifted despite being hard to enact in practice and, at times, certified as being unlawful. Rather, an endless stream of pretexts to maintain priorities as they were was produced using available data, justifications or themes that gain traction. The relationship between state power(s) and this policy field can be gauged from the contents, definitions and envisaged effects of legislative acts, starting from the national level and also at the EU level from Part 2. A serial production of hierarchies to enhance state power(s) and dynamics of expansive exclusion is evident. The relational concept of power in this model excludes the existence of "power in general", preferring "to establish the weight of different sets of particular powers and how they combine, if at all, to produce specific structures of domination" [3].

After drawing on Bourdieu's insights to explain why an overt shift to necropolitics may undermine the credibility states draw their authority from and further illustrating the SRA's theoretical bases and their relevance to current developments, the next section analyses a "fix" that pits immigration policy against normative frameworks.

## 4. Making Sense of the European (EU and MSs) Migration Policy Fix

At the EU level, the focus since the mid-1990s was to constitute power(s) and security structures under the Schengen AFSJ. Initial frameworks sought to drive the expansive development of EU immigration and security apparatuses, setting overambitious goals in these fields to guarantee long-term expansive activities and enhanced powers to wield. Evaluation mechanisms were provided, but the 2005 Hague Programme (PP2, 2005, Part 3) required that they be limited to verifying works without assessing the validity of agreed approaches or underlying premises, to propose improvements to existing practices and goal attainment. For two decades, migration policy has expansively promoted large-scale human rights violations at the EU and intergovernmental levels. Such violations are committed by MSs or their agents deployed in EU actions, and a restrictive outlook fuels discriminatory practices and discourse within and beyond EU borders. A state power perspective and a long-term document-based approach focusing on hierarchy, strategic selectivities and structures help to clarify what happens beneath complex inter-level dynamics.

From 2014, a mutation occurred after this policy field became more structured in Parts 3 and 4 (2005–2014). From a power grab by executive authorities at different levels, it morphed into a wrecking ball in relation to normative frameworks, from the Geneva Convention on Refugees to the UN Convention on the Rights of the Child (UNCRC), the law of the sea, national laws and constitutions, the European Charter of Fundamental Rights and Freedoms (ECFR) and the ECHR. This was most evident in "frontline states" for which there is abundant literature on events at sea and conditions in hotspots, but northern MSs did not escape such dynamics (as Canning notes concerning the degradation of reception conditions for asylum seekers "by design" in the UK, Sweden and Denmark, [28]). Significantly for a policy field whose declared purpose was to protect EU citizens' security and rights, several EU citizens were criminalised if they engaged in acts of solidarity towards migrants and refugees [29–31].

Dynamics leading to the portrayal of human rights and any obstacle they pose for state actions against irregularised migrants as "problems" in Parts 3 and 4 continued along an expansive vein followed to normalise discrimination and subject TCNs to differential treatment often amounting to abuse in Parts 1 and 2. What seem to be aberrations enacted since 2014 were planned, agreed to and implemented through a series of policy initiatives

and executive acts (by MS governments) whose failure and detrimental effects never led to a rethink. A strategic outlook was developed through early informal intergovernmental cooperation that hardwired a security-minded approach to migration into the EU JHA policy field (PP1). In time, this led to disregard for the problems this approach caused, alongside instrumental use of developments within and beyond the JHA field to intensify policies against migrants and mobility. Terrorist attacks in Part 2, like those in Madrid and London in 2004 and 2005, ushered in demands to "terror-proof" immigration and asylum policy and add an "external dimension" to EU JHA policy in Parts 2 and 3 (EUD10; EUD16).

The 2008 Pact on Migration and Asylum (EUD19, Part 3) was agreed to by the Council to uphold the EU migration management model's strict outlook opposed to large-scale regularisations like those enacted in Italy (2002) and Spain (2005) [32]. EUD19 exemplified efforts to stop MS practices disregarding the general outlook of EU policy for pragmatic "government" reasons. The new Pact proposed in 2020 has a similar function, to set in stone unlawful practices enacted under the 2015 European Agenda on Migration in exchange for minor adjustments to the overall system that cannot be postponed. A possible "brain drain" from countries of origin and funding required for social protection to maintain the direction of travel were evoked by a Pact that disregards the cost and social harm caused by criminalising unauthorised mobility. The model in question requires differential treatment for third country nationals (TCNs hereafter) and coercive measures to limit mobility. A 1995 precursor of the EU visa list (EUD1, Part 1) of states whose citizens require visas to travel featured the bulk of countries ranked lowest in the previous UN Human Development Index [33], with a few exceptions (46 of the bottom 50 countries and 75 of the bottom 100 were on the list). Such evidence highlights that this policy field operates to entrench global inequality by collectively portraying states' entire populations as a "*migration risk*", which may become a "*migration and security risk*" if countries experience problems beyond poverty, like armed conflict. As consistently emerges from the documentation, states denying TCNs their individuality, rights and capabilities through legal and strategic means has been the discursive, substantive and structural core of immigration policy. Frontex provided a means to promote measures against migrants (and inherently discriminatory practices) as a technical and scientific exercise driven by data collection, risk analysis and the serial production of distinctions and hierarchies between population groups. MS governments hostile to migrants may welcome advice to introduce tough measures and treat Frontex's recommendations as authoritative when framing their interventions.

The aim of expelling people residing irregularly in the EU territory was codified and regulated at the supranational level in Part 3 (EUD20). As this policy field matured, the European Parliament (EP) used the co-decision procedure to demand limited human rights safeguards alongside a duty for MSs to systematically issue return decisions. Five years later (EUD25), at the Council level, scant consideration was given to cases of abuse, self-harm, violence and irregularities affecting migrants in detention. Human rights safeguards were deemed a problem that needed solving to "restore the credibility of the EU returns system". The use of aggregated figures on expulsion orders issued to back this claim disregards that people sometimes receive several expulsion orders, and they are issued to people who cannot be expelled in violation of normative frameworks. Thus, the situation may be portrayed as one of crisis, which entails systemic effects. In a similar way, projects like Eurosur (EUD24, Part 4) mark a shift towards a hi-tech military conception of borders for which the incremental securitization promoted by Frontex since it began operating is inevitable, disregarding its ethical, financial and normative costs. This outcome is achieved by detaching the effective pursuit of strategic goals, supranational governance and inter-governmental steering from what they mean in practice. EU institutions and agencies are withdrawn from accountability due to a lack of operative capability on the ground: Frontex, the Commission or the Council may recommend and engineer violations of people's rights and rules to be enacted by MSs, but they are not responsible for such breaches.

After outlining why EU and MS immigration policies may be incompatible with human rights, document analysis continues below, showing how the EU state-building project is being misused at different levels for instrumental purposes.

## 5. EU Development as a Long Transitional Phase—From Power Grab to Wrecking Ball

Structural, motivational and ideological aspects seldom receive the attention they deserve. They are hard to capture with precision despite underlying the EU's essence as an evolving transformative state project in continuous transition. Gramsci's [12] explanation of "*the interregnum*" as a time when the old order has not died but the new one struggles to be born, portrays it as a time when all kinds of morbid practices materialise. In the EU JHA policy field, an initial intent to develop EU security structures set up expansive dynamics for policy intervention and enforcement, creating structures and spending resources on technological solutions. This unbalanced outlook has not been remedied. The opposite seems to be happening, as a drive for effectiveness consolidates practices and legal norms initially deemed exceptional or plainly unlawful but necessary in the short term. The ideological aspect mentioned above goes some way towards explaining these dynamics, as documents in Part 1 that framed JHA aspects of EU immigration policy showed.

Documents EUD2–6 stress a need to discipline societies and public service workers to act in an uncompromising fashion towards irregularised migrants. EUD1 set a tentative (later withdrawn and replaced) list of countries whose citizens must possess valid visas not to be deemed irregular and treated accordingly (an epitome of hierarchy applied collectively to people from certain third countries). The ideological dimension of EUD2 was stark: It recommends that MSs inform central and local authorities involved in service provision "*of the importance of combating illegal immigration*" (point 4). Proactive reporting of breaches must be encouraged, the attention of authorities issuing residence permits should be drawn "*to the risk of marriages of convenience*" (premise 4) and targets of migration policy-related penalties incorporate employers who may incur fines and criminal charges for profiteering or not complying with control duties. The documentation shows that a wave of repression and control unleashed in the guise of migration policy would not just affect its supposed targets, nor be limited to borders. Different categories were drawn into this fight, starting from commercial carriers and their staff, whose interests were subordinated to migration policy enforcement. This coercive effort also harms people's security, including that of EU (and UK, Griffiths) [34,35] nationals, if they have family ties with TCNs and if their physical traits are not associated with European identities. Such features may give rise to the suspicion that someone's presence in a territory may be irregular.

Two justifications for EU and MS migration policy and restrictive migration policy regimes, more generally, are that states can choose who to let into their territory, alongside a theoretical defence of compatriot favouritism [36]. Despite convincing philosophical challenges to this outlook on grounds of ethics and fairness [37,38], scope to challenge this policy field as it exists and operates may also focus on substantive aspects like its detrimental effects for people and states it supposedly protects, and its effects beyond EU borders. This research highlights an inherent relationship between this policy field and drift towards "authoritarian statism" [19] written into its material and strategic development. Despite being hamstrung by reliance on high-level documents (rather than those from technical, operative or bureaucratic milieux, or observing events on the ground), such aspects emerge with greater clarity than was expected. The 1999 Tampere Council conclusions (PP1) formally began the development of the AFSJ by merging two separate policy fields (immigration and asylum) and incorporating them into JHA policy.

Conflating migration and asylum follows a blueprint whereby MS governments used the EU Council to devise strategies and authorise each other, individually and collectively, to violate normative frameworks and limits to state power(s) to pursue policy goals. Documents in Part 1 offer evidence of this, through explicit accounts of how states might relinquish their duties and arrogate enhanced powers. Hence, minors should be deportable despite their best interest being prioritised by the UNCRC. A lack of documents to establish

identity may be circumvented one-sidedly using an EU *laissez-passer* document to enact deportations. Unaccompanied minors who are asylum seekers and over 16 years of age may be placed in reception facilities with adults (EUD2–4; NLIT1, Part 1, 1995–1999). The serial production of hierarchies also applied to children: First between EU nationals and those from third countries; then between regular migrants and illegal entrants; then between refugees and "economic migrants"; between accompanied and unaccompanied minors; and, finally, between over-16s and under-16s. As a protected category, asylum seekers were used to criminalise economic migrants before this policy field's expansive outlook focused on denying them access to asylum and the enjoyment of decent reception conditions that they are due under international and EU law. MSs gradually acted to conceptually turn refugees fleeing conflicts or repression into illegal entrants, in unison with EU efforts.

Countless attempts to overwhelm rules and undermine human rights must not distract from systemic and structural effects. Seeking to assert states' power over human mobility (in a biopolitical key) offered a gateway for racism and discrimination to become core features of EU security structures. This belies EU claims of compliance with, and promotion of, human rights and international law. As time passed, this policy field's outlook sought to exclude irregular migrants from humanity with expansive social effects [39], and to affirm human rights as differential privileges [40] that only apply to citizens and legal residents (criteria to deny people legal status and acquired rights expanded). NLUK4 (Part 4, 2010–2014) was the epitome of this outlook. This law aimed to create a "*hostile environment*" for illegal migrants and stated that "*a private life established by a person at a time when the person's immigration status is precarious*" should carry "*little weight*" in decisions on removal. Considering the Black Lives Matter protests that followed the killing of George Floyd in 2020, it is hard to find clearer examples of legislation that codifies the opposite view to this rallying cry.

A choice appears to have been made in 2014 that human rights and the rule of law impede correct implementation of the EU's migration management model, leading to their structural subordination to achieving strategic migration policy goals. Three such goals amount to reducing irregular border crossings into the EU and asylum applications submitted in MSs, while increasing the number of returns (deportations) to "restore the credibility of the EU return system". The dogged pursuit of such goals over time entails large-scale human rights violations in practice, but these phenomena are conveniently decoupled to protect Frontex and the EU from bearing responsibility for unlawful procedures and practices they promote. Later events seemed aberrations (whether they regard letting people die at sea, normalisation of mistreatment camps outside EU borders or undermining asylum seekers' reception conditions and access to asylum procedures on Greek islands [41]), but they comply with the blueprint and strategy pursued from 1995 to 2014. Undermining international law and enhancing state power over people through expansive modes of discrimination starting from nationality (added to by further concerns, serially raised and codified into laws, from income to evaluation of character or risk profiling) belongs in this policy field's substantive core.

Visvizi et al. [42] note the instrumental use of migration in public debate, including in EU countries, and the different perspectives that must be considered to conceive a shift from national sovereignty-based management to a collaborative global governance model. They stress the present incoherence of "a patchwork of loosely, if at all, connected systems and ad hoc solutions" and "wasted opportunities of synergy-building" as exacerbating a "mismatch" between a "Westphalian" management model and the phenomenon of migration as its negation. Tentative explanations for global governance remaining a "distant ideal" (the question they pose) may lie in their insight that international law sets states' obligations regarding "refugee law, humanitarian law and human rights law" and in a need for unbiased, non-hierarchical, dialogue.

Below, the document analysis focuses on three elements of the research design: hierarchy, strategic selectivities and structures.

## 6. Hierarchy

The first point to note about hierarchy is that perfecting an overambitious migration management model has been a core JHA policy strategic goal for 20 years without adequate justification and despite plentiful evidence that it is counterproductive, not just for migrants. Secondly, the safeguards that EU policies and laws are not allowed to disregard are human rights and protections due to vulnerable categories or those protected by international law and rights frameworks. On inspection, this policy field can bluntly be typified as a power grab to undermine limits to what states can do to people. In practice, it has been a long-term onslaught against TCNs, asylum seekers, families and minors, unaccompanied minors, vulnerable subjects and refugees.

Hierarchy proved useful to reveal the sequence of phases and priorities used in pursuit of expansive migration policy enforcement, providing insight into the dynamics at work. The research methodology linked hierarchy to JHA aspects of migration policy, but hierarchy also emerged in other ways, some of them unexpected. In Phase 1 (1995–1999), a relatively "new" immigration country (Italy) was defining an EU-compliant policy to assist asylum seekers and constitute irregular migration as a problem to tackle. Lawmakers sought to respect constitutional principles (on asylum, non-discrimination, etc.) despite inherent risks posed by practices of criminalisation, detention, deportation and exclusion. The codification of hierarchical relations in Italy (NLIT1) was matched by expansive exclusion and control in the UK regarding meagre entitlements afforded to semi-destitute asylum seekers and to deny them a choice of destination or place of residence (NLUK1, NLUK2). Enhanced sanction regimes, controls and the possibility of incurring sanctions for not applying immigration law correctly were affecting other categories, subordinating their commercial interests and/or personal ethics. Similarly, a focus on borders and who to allow into the country spread inwards, focusing on expulsions and whether expellable categories can include people who had established their right to stay or, later, even UK citizens whose status may be revoked.

Hierarchy features in documents through the continuity with which actions against irregular migration remained priority areas for intervention over time. Hierarchy influenced the actors who could intervene and influence proceedings in an intergovernmental (Council-led) policy field, closely bound to rights, at a time when security-minded outlooks were in the ascendancy among national governments (1995–2014). The conversion of centre-left political parties to an outlook previously associated with the right or centre-right was crucial in this phase ([43], on Italy; [44], on France; [45,46], on the UK). In Phases 2 and 3 (2000–2009), as a precondition for adhesion, countries that joined the EU in different rounds between 2002 and 2009 adopted and began implementing the JHA *acquis* in full, without having a say on matters of constitutional importance. Italian legislation (NLIT2) headed towards a crude structural subordination of migrants, assigned responsibilities to employers to discourage them from recruiting non-EU workers and intensified power differentials between them and TCN employees. The UK shifted from devising ways to exclude people who had already acquired rights towards stripping its nationals of citizenship for expulsion purposes, or to making it hard for them to live with foreign spouses unless their income was high enough. Income followed nationality as a valid reason to undermine people's lives in the UK, after which a legal provision in Phase 4 (2010–2014) stated that people's lives and activity while their immigration status was unclear should carry scant weight in decisions about removal. Meissner [47] notes that frequent rule changes in the UK affect legal status differentiation regimes, stressing how the regulation of migration is a driver of diversity. Such differentiation is viewed as a means of ordering migration to optimise outcomes, and as a preserve of national approaches, as opposed to a globally interconnected [or supranational] policy field. Interpreting different tracks to gain legal status, conditionalities of entry and parameters of presence by using gap-filling narratives neglects the social and economic implications of status differentiation, patterns of marginalisation and the "social inequalities status track differentiations entail and obscure".

A significant hierarchical aspect to understand dynamics in EU migration policy is that state actions are formally subordinated to human rights and international law (particularly in cases involving minors, refugees and protected categories). This led MS governments to use the Council as an arena in which to authorise each other to undermine normative frameworks in pursuit of effective migration management. Unlawful practices thus introduced through soft law enabling instruments like Council conclusions would then be peddled as new applicable rules in negotiations with the Commission and European Parliament after 2009 and the entry into force of the Lisbon Treaty (including secret negotiations, "trilogues").

Three hierarchical aspects maintained over time help to understand inherent features of EU migration policy. The first one is that states should be able to exercise unbounded power in relation to people. The second aspect is that EU and MS interests must condition the conduct and internal policies of third countries, without concern for harmful effects such endeavours may inflict on their people (including those who have migrated) and societies. A third hierarchical aspect is that MSs must comply with EU policy steering and harmonisation, at the same time as state governments acting individually and collectively within the Council dominated the JHA policy field. This context explains the failure to improve human rights standards, checks and balances. This field's intergovernmental bias led each MS to protect anomalous or legally questionable practices used by their law enforcement agencies in policy enforcement on the ground. The EU (supranational, intergovernmental) and national levels shared an interest in maintaining existing powers and competences unaffected, while new powers and structures were constituted to enhance and expand them.

Over time, new proposals were made but the priority accorded to fighting irregular migration as a precondition, even in discussions about opening legal migration routes, remained a constant feature. Discussions in the Council in Part 4 (2010–2014, EUD24 and 25) amounted to subordinating human rights safeguards and the rule of law to achieving strategic migration goals—more detention and returns, fewer arrivals, fewer asylum applications, swift exclusion from asylum procedures, more fingerprinting, fewer SAR activities and more aerial surveillance. Similar arguments in Part 2 (2000–2004) were justified using antiterrorism and the need to "terror-proof" asylum and immigration policies; in Part 3 (2005–2010), PP2 indicates the emergence of a parallel "*Raison d'Etat*" at the EU level, due to the priority given to developing the AFSJ, with immigration policy at its core. The creation of Frontex late in Part 2 marked the moment when an agency was tasked with promoting its role as a vehicle to steer state actions towards any practice conducive to effective migration policy enforcement. Analysing its work, the instrumental use of data to promote systemic discrimination, disregard for people's rights and limits to state powers to objectify people in order to advance policy is evident. Frontex deploys an alarmist narrative to portray irregularised migration as an unsolvable existential threat without investment in controls and coercion, promoting racism and human rights violations within and beyond EU borders, and weakening international law. EUD15 (the 2004 Frontex Regulation) tasks it with asserting its role and capabilities to develop an integrated control and surveillance system at external borders to gradually improve "external border management", including through an effective return system. More generally, it engaged in an ideological struggle to establish that migration management is a valid reason to undermine positive values, goals and foundational principles that the EU supposedly promotes, from the rule of law to the right to information and due procedures. A hierarchical aspect requires that MSs should not obstruct the agency's activities:

> "*Member States shall refrain from any activity which could jeopardise the functioning of the Agency or the attainment of its objectives*". (Art. 2(2))

Early in Part 2, the 2000 EC/ACP Partnership Agreement (EUD8) was used by the EC to make development aid and support for UN Development Goals conditional on readmission. A duty to readmit (reciprocal, although it was about EU states deporting people, as is obvious in the text) was codified in art. 13.4 of the Agreement, repeatedly

cited in EU migration policy documents up to the present day as a legal basis allowing the EU to deport irregularised residents to any country of origin party to the agreement.

## 7. Strategic Selectivities

Identifying strategic selectivities by tracking dynamics in governance that emerge from policy documents reveals a systemic imbalance maintained for the period under scrutiny. It amounted to migration management, border controls, constituting EU security structures and the pursuit of strategic policy goals prevailing over respect for human rights, international law, principles like non-discrimination and positive values like the rule of law and ethics. The priority given to perfecting EU migration policy was regularly reaffirmed to prevent let-ups or decreases in intensity, harnessing crises and data conducive to this goal. Migration control was prioritised over compliance with human rights in the ECHR and ECFR, legal safeguards in national constitutions and in international conventions and normative frameworks. The 1950 Geneva Convention on Refugees, the law of the sea, the right to life and the right to personal freedom have been compromised in different ways, within, and more severely outside, EU territory. Commitments and duties under the UNCRC and the Convention against Torture (CAT), including non-refoulement, have been relinquished *de facto*, despite formal claims of compliance in the preliminary observations of policy documents.

Structural sequencing offers evidence of strategic selectivities. When a need to open legal pathways and create safe forms of travel is mentioned, it is subordinated and sequentially ordered to come after irregular migration has been stopped. Setting arbitrary, ideological and overambitious targets creates a context in which possible changes of direction are effectively blocked. Broadly, the documentation makes it possible to conceptualise a framing phase (harmonisation) in Part 1 (1995–1999) to promote expansiveness and common standards, a narrative portraying migration as an existential threat, and ways to circumvent limits to states' lawful exercise of power(s). In Part 2, an antiterrorist emergency was used to justify externalising migration policy and portraying migrants as threats. A Commission Communication on "a common policy on illegal migration" unveiled in November 2001 (EUD10) reaffirms:

> "The prevention and the fight against illegal migration are essential parts of the common and comprehensive asylum and immigration policy of the EU".

Specification of a red line that must not be crossed, written in bold for emphasis, reads:

> "Illegal entry or residence should not lead to the desired stable form of residence."

These statements document an institutional outlook geared at turning migration into an intractable problem, ensuring that people who enter "irregularly" (many of whom could not enter otherwise) can only be a "burden" for host societies, regardless of conduct and personal qualities. Failing to provide pathways for regularisation may be a problem that helps maintain irregular migration as a strategic priority. Vassallo Paleologo [48] argues that a far-reaching regularisation procedure is needed to allow a hidden population that is forced underground by immigration policy enforcement and irregular status to emerge and make a positive contribution. From the EU governance outlook embodied by Frontex, this would be a "pull factor" for irregular migrants, as applies to people who help migrants, SAR operations at sea and, increasingly, dignified refugee reception conditions.

Political agreement to create a permanent agency to promote immigration policy enforcement at EU external borders placed an expansive drive to criminalise migrants and mobility at the core of JHA policy, including external action for home affairs policy purposes. The latter strategy undermines the theoretical justification of a country or supranational entity's right to decide who to allow into their territory and "compatriot preference" [36], because these strategies harm people and societies in third countries. Policy-steering efforts strengthened this tendency (EUD16 on the external projection of the AFSJ, the Global Approach to Migration, EUD17) to project harmonised measures

abroad (develop a "coherent external dimension") in countries that may disagree and do not benefit from EU membership:

> *"The projection of the values underpinning the area of freedom, security and justice is essential in order to safeguard the internal security of the EU".*

This outlook was reaffirmed in policy-steering documents in Part 3 (2005–2009), most evidently PP2 (The Hague programme) and EUD19 (European Pact on Asylum and Immigration). In the first case, an overarching security and antiterrorist outlook was treated as a durable structural feature by referring to border controls, internal security and the prevention of terrorism as "indivisible". Hence, improving state capabilities to "regulate migration flows", "control the external borders of the Union" and "fight organised cross-border crime and repress the threat of terrorism" is deemed coherent with improving capabilities to "guarantee fundamental rights" and refugee protection. This narrative disregards the link between human rights violations and law enforcement, migration management and border control activities (certified by courts including the ECtHR), nor does it consider that upholding human rights requires states to stop violating them. Externalisation brings up the issue of funding or influencing third parties to commit human rights violations in the guise of cooperation to enhance state capabilities for border control and migration management [49].

The "limited resources" argument (a recurring pretext to subordinate expanding legal migration options to effective migration controls) used in EUD19 and EUD17 to justify continuity and expansiveness is undone by the cost and aims of actions deployed against irregularised migrants and mobility. This outlook spreads in Part 4 (2010–2014) to "governance" strategies that undermine rights and legal safeguards to "restore the credibility of the EU return system" (EUD25 and EUD26). Expansion is deemed an inherently positive value and these policies' detrimental impact for rights within and beyond EU borders is entirely disregarded, as are valid reasons third states may have not to strike readmission deals or allow implementation of EU borders in their territory. The substance of this policy field appears to be that borders must be impossible to cross for people without authorisations that a portion of the world population cannot obtain, but they should not matter for immigration policy enforcement.

The evaluation process may contribute to change based on the shortcomings of strategies deployed in the fight against irregular migration. However, from early in Part 3 (2005–2009, PP2), it was deliberately left toothless. Evaluation should not pose "too heavy an administrative burden", nor should it challenge measures adopted, underlying premises or rationales. Rather, its goal

> *"should be to address the functioning of the measure and to suggest solutions encountered in its implementation and/or application"* (p. 2).

Continuity based on an idea that progress must not be undone by questioning structural flaws and/or underlying premises was matched by keeping the same strategic goals over time, even if they prove hard to achieve and provoke resistance or opposition. This applies to the emphasis placed on striking readmission agreements with third countries from 1995 to 2014, reiterated in 2015 (EAM) and 2020 (draft Pact on Asylum and Immigration), and to a drive to detain and deport ("return") in which Frontex plays a leading role. Recasting the Returns Directive (EUD20) and reviewing the Frontex Regulation expansively for the umpteenth time are deemed crucial passages to restore "the credibility of the EU return system", heralding a phase of operations to find and deport irregularised migrants throughout EU territory.

## 8. Structures

The creation, development and operation of structures through legislative acts as instances of "government" illustrates aspects of EU and MS migration policy that advance, producing concrete outcomes, and others on which progress is slow, sequentially subordinated and hard to achieve. It is noteworthy that government structures to control or fight

migration (and new modes of intervention, practices and procedures) are swiftly created and become operative. Instead, anything that may improve the plight of migrants and asylum seekers or safeguard rights is cited prospectively, whereas implementation tends to be delayed. In the first category, we find EURODAC and Frontex at the start and end of Part 2 (2000–2004) and EUROSUR in Part 4 (2010–2014), alongside structural items with systemic relevance about visa lists, punishment for carriers and facilitation, an assertion of the duty to expel and related aspects or the Schengen Borders Code. In the second category, we see the creation of a Common European Asylum System (CEAS) on which progress has been slow, as is true of a common visa management system to be managed through MS consulates. EASO (European Asylum Support Office) was created to facilitate access to asylum procedures. However, this agency was not fully developed and operative until it was flanked in the implementation of hotspots and the EAM in 2015 by Frontex, to which it was made subordinate, as mechanisms to exclude asylum seekers from procedures (2016 EU-Turkey deal) were devised.

Apart from EU legislative instruments, the value of focusing on MS case studies shone through in the analysis of national legislation in a country (the UK) that was a forerunner and inspiration for the EU migration policy model, and in another one (Italy) that did not have an established system. In the latter case, in Part 1, NLIT1 shows the difficulty of enacting policies from the EU level in compliance with a Constitution that offers appropriate human rights protection, forbids discrimination and interprets the right to asylum widely. On the contrary, NLUK1 and NLUK2 belied claims that this policy field was about regulating entry at the border, as practices strayed towards undermining protection and support for refugees and providing grounds to disqualify people who enjoyed acquired rights (through notions of criminality, misbehaviour and character flaws). This policy field's inherent tendency to produce hierarchies between population groups was evident in the various types of citizenship that apply in an "old" migration model like the UK. Likewise, a drive to strip people of acquired rights and to make them conditional confirmed this policy field's expansive outlook, underscored by measures to be required of third states.

National legislation lends materiality to EU migration policy-steering documents, Directives, Regulations and Framework Decisions because it is closer to the point of delivery, as acts of "government" valid in specific jurisdictions to be applied in practice. This is not to disregard the fact that laws may be challenged and are not always fully implemented, but they represent a step towards concrete implementation of policy. Laws' contents show what adjustments are necessary to coordinate EU and national normative acts and how instructions drawn from supranational "governance" translate into legal measures. In concrete terms, while at the EU level the Eurodac database was set up to operationalise the Dublin Regulation system, at the national level (NLIT1), guarantees were introduced to accommodate the Italian Constitution's prohibition of discrimination and ontological reading of the right to seek asylum. A plethora of databases was needed for several purposes, including to ensure that TCNs would not take jobs from "natives", for which NLIT2 required that employers check the availability of Italians before seeking to employ outsiders. The same law provided measures to expand treatment of "facilitation" (blurring distinctions between smuggling and trafficking) as organised crime, for which Italy has understandably tough laws, investigative and accessory measures thought up for Mafia and Mafia-like syndicates, whose use has spread to citizens acting in solidarity with migrants. This is one of the countless examples of definitions being unduly stretched to accommodate strategic migration policy goals.

In Parts 3 and 4 (2005–2014), legislative measures at the EU level matured without resolving their underlying flaws, while national laws turned towards devising ways to detain and deport supposedly protected categories like EU nationals. As deportation and tracking down "irregular" migrants became a priority, it became less important who was being deported, as transpires from the documentation that worked towards disqualifying people from rights acquired through citizenship (NLUK3 and NLUK4). This drift became

obvious in the later Windrush scandal in the UK [46] that affected people who entered the country legally many decades ago. At a supranational level, the Returns Directive (EUD20) crystallised two decades of provisions through which MSs authorised each other in the Council to violate human rights to achieve strategic migration policy goals. The Schengen Borders Code also marked a switch of the EU's influence from governance to government, by setting out rules to be homogeneously applied at every EU external border (sea, land and air) whose main feature was to institutionalise discrimination between EU citizens and TCNs.

These pieces of EU legislation provide loopholes that allow MSs some discretion when applying EU measures. There were examples in national legislation (NLIT3, NLIT4) of implementation that exploited such exceptions to avoid applying more favourable measures in EUD20, for instance by criminalising unlawful entry and stretching permitted detention limits using available exceptions. Likewise, when the Lisbon Treaty brought the JHA policy field more firmly into the EU system (allowing the ECJ to monitor compliance with EU law), a standing committee on internal security (COSI) was created to continue intergovernmental initiatives led by interior ministries and law enforcement. This flight from accountability also involves moving problematic interventions from the JHA field to those of the European External Action Service (EEAS) and military-led CDSP (Common Defence and Security Policy) missions. Diplomatic and operational concerns mean that information becomes easier to keep under wraps, a trend that worsened after 2014. An archaeological approach and switching between EU legislative acts and national laws lend substance to EU measures by providing rules for implementation and addressing aspects deemed to need correction. Thus, early measures like Regulations concerning Eurodac and its mode of operation did not envisage that people would resist fingerprinting but, as time passed, evidence shows that people resisting fingerprinting became another problem tackled through coercive legal measures.

## 9. Conclusions: Immigration Policy as State Power

This article employs a methodology reliant on state theory that includes concrete elements and draws a distinction between formal and substantive aspects of European migration policies. This distinction allows inherent contradictions and distance between this policy field's claims and substance to emerge [4]. Formal recognition of human rights' hierarchical prevalence is included in documents that devise, propose or formalise strategies to neutralise their effects in practice. A focus on state power(s), strategic selectivities and structures leads to conclusive evidence about structural imbalances and dynamics that fulfil the goal of Jessop's SRA [3] to provide

*"a certain apparatus and operational unity horizontally and vertically"*.

Emphatic answers are provided to four research questions about expansiveness, why and how this policy field has strayed beyond its remit, the relation between identified shifts and authoritarianism and whether migration policy is more about creating problems than solving them.

(1) Immigration policies have exceeded their ostensible scope geographically and through expansive development of procedures and criteria to exclude and disqualify people, even negatively affecting people who are not migrants. (2) The expansive exercise of coercive state power(s) is driven by structurally inscribed strategic selectivities that fail to consider that there may be a point beyond which such powers come to represent a problem and disregard their effects in third countries. (3) A proliferation of grounds to justify controls and the adoption of punitive measures increases power differentials between state agents and people upon whom differential regimes are imposed, and among people throughout society. (4) Setting overambitious policy goals leads to problems that require normative frameworks, competing positive values and societal standards to be overwhelmed to achieve arbitrary goals (reworked from [4]).

This article's introduction outlines a study to identify inherent features of JHA aspects of immigration policy from an SRA perspective framed around hierarchy (and

state power(s)), governance (and strategic selectivities) and government (and structures). Section 2 questions the reasons for immigration policy's expansive development, excesses and illegal practices, introduces the SRA and proposes content analysis using state power(s) as an interpretative key. Section 3 draws on Bourdieu's insights about the state's need to appear credible to operate effectively without drifting towards imposition and tyranny. It outlines useful aspects of the SRA for researching immigration policy, its theoretical pillars and uses EU governance and Frontex as examples of "multiscalar metagovernance" and "depoliticalisation". The discussions in Sections 4 and 5 include findings from policy document selection that lend substance to the theoretical analysis that they accompany. Section 4 focuses on the inherent incompatibility between immigration policy as it currently operates, international normative frameworks (including human rights that are supposedly binding for the EU and MSs) and the rule of law. It argues that the EU may be caught in a policy fix. Section 5 sounds an alarm, because a long transitional state-building phase may be a gateway to assert approaches amounting to a wave of coercion and violence unleashed on society, starting from, but not limited to, TCNs and irregularised migrants. Sections 6–8 present significant findings relevant to the hierarchy (6), strategic selectivities (7) and structures (8). Hierarchical aspects include structurally subordinating competing interests to migration policy objectives and introducing distinctions between population groups that are the core of this policy field. The key strategic selectivity in the documentation is an inherently expansive vocation, but it is as important to notice what discussions and paths for change are shut down, and how. Structures and their development phases are the embodiment of intention, strategy and the "institutional materiality" of a policy field.

Plentiful research focuses on this policy field's inherently problematic societal aspects [50] as a means to entrench global inequality, influencing third countries through the externalisation of a policy field laden with violence and coercion that has colonial roots and overtones [51,52]. This methodology and theoretical outlook contribute to shedding light on the institutional workings behind the normalisation of cruelty, state crimes [11] and human rights abuses on land and at sea, within and beyond the EU's borders. This policy field is driving a developmental regression [53] that degrades democratic and human rights standards within the EU, causing a proliferation of "ignored insecurities" sometimes linked to policy enforcement [54]. This problem is compounded by influencing third states, to achieve policy goals that drive their authorities towards discrimination and contempt for limits to their power(s) in pursuit of effectiveness. Jessop's notion of a "state project" helps to make sense of developments in statehood. Alongside a supranational governance model to develop JHA structures through the AFSJ, it may be viewed as providing authority to overwhelm limits to state power(s), establishing a parallel "Raison d'État" to that which exists at the national level. One may hope that highlighting structural flaws and systemic imbalances in this policy field that have not been fixed for over 20 years may help to plot a path out of the migration policy fix.

Thanks are due to Tony Bunyan (Director Emeritus of Statewatch) for pointing me towards Bob Jessop's work to make sense of the EU as a state project, to Ann Singleton and Christina Pantazis for supervising my PhD, to the editorial team at Societies and to three anonymous reviewers for their comments, support, advice and time.

**Funding:** This paper received no external funding. However, it relies and builds upon a PhD thesis, awarded in November 2019, funded by an ESRC collaborative studentship (Contract 41416) facilitated by SWDTP, the University of Bristol School for Policy Studies and Statewatch.

**Institutional Review Board Statement:** Not applicable.

**Informed Consent Statement:** Not applicable.

**Data Availability Statement:** The full list of empirical documents and links has been uploaded in a dedicated section of the Statewatch website at https://www.statewatch.org/observatories/immigration-and-asylum-in-europe/thesis-on-migration-policy-and-state-power-selection-of-empirical-documents/ (accessed on 24 October 2021).



**Conflicts of Interest:** The author declares no conflict of interest.

## List of Acronyms

| | |
|---|---|
| AFSJ | (Schengen) Area of Freedom: Security and Justice |
| CAT | (UN) Convention against Torture |
| CEAS | Common European Asylum System |
| COSI | Standing Committee on operational cooperation on Internal Security |
| CDSP | Common Defence and Security Policy |
| EAM | European Agenda on Migration |
| EASO | European Asylum Support Office |
| EC/ACP | European Community/African, Caribbean and Pacific countries |
| ECHR | European Convention on Human Rights |
| ECtHR | European Court of Human Rights |
| ECFR | European Charter of Fundamental Rights and Freedoms |
| ECJ | European Court of Justice |
| EEAS | European External Action Service |
| EU | European Union |
| JHA | Justice and Home Affairs |
| MS | Member State |
| PoS | Place of Safety (as conceptualised in the law of the sea) |
| SRA | Strategic-Relational Approach (Jessop) |
| SOLAS | International Convention for the Safety of Life at Sea |
| TCN | third-country national |
| TNI | Transnational Institute (Amsterdam) |
| UN | United Nations |
| UNCRC | United Nations Convention on the Rights of the Child |

## Notes

[1] Indicates translation by the author from the original language version into English.

[2] The author prefers this term to "irregular" because it foregrounds institutional activity undertaken to undermine people's status on account of unauthorised mobility.

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
