# Peer review of "Immigration Policy and State Power"

_societies, doi:10.3390/soc11040128_

Round 1

Reviewer 1 Report

The article does not follow the instructions for the authors as the journal required. There are some incoherencies in the text, e.g. Part 1-4 (line 83).

Author Response

Dear Reviewer:

Thank you very much for reading and commenting on my paper. I have fixed the reported incoherence, specifying the time frame for each Part and that they refer to the tables of selected empirical documents for each five-year period (new lines 32-33).

I have sought out and reviewed further points or formatting that may have been unclear, as part of an overall copy editing exercise. I have also provided more details on the methodology, empirical materials and research strategy in chapters 2, 3 and 4.

I had reasons for not keeping to the format and specifications because this is experimental work on historical documentation to uncover dynamics that result in current systemic malfunctions. The foci are theory and methodology to uncover why the shifts in model or phase raised in the CfP appear inevitable. Changes advance in the guise of novelties and may sometimes seem aberrations, but they have long dwelt in plans against irregularised migration as a standard feature of this policy field. The theorised shift from biopolitics to necropolitics does not refer to a goal or unavoidable outcome of policy as death (which dates back to much earlier), but rather, to this choice becoming explicit since 2014/2015. Strategies of justification, blame deflection and demonisation of recalcitrant actors have since been deployed to justify death, or causing or leaving people to die in defence of EU borders. A parallel “Raison d’Etat” to the one that exists at the national level appears to be developing at a supranational level to impose policies that are detrimental to the EU by promoting selfishness (perceived national self-interest) in the midst of an ambitious and (formerly) sophisticated collaborative state-building exercise. I improved my compliance with the instructions by laying out a clearer introduction and conclusion, shortening both and making them more instrumental for the article’s purposes and as guidance for its readers. I also added signposting in the individual section and methodological notes. This has also been in response to my article’s other two reviewers, who requested these changes and to remove references to my PhD, which was another feature of the article that made it unorthodox. This has freed up space to address the editors’ concerns more closely, primarily near the start of section 2 (l. 88-141) and thereafter. These items deal with the intersection between Frontex’s analytical and advisory roles that bridge the national, supranational and intergovernmental levels with a strong ideological impact, and the normalisation of problematic practices as “effectiveness” in the pursuit of policy goals. Alarmism and overreaction are built into the adopted model for the Justice and Home Affairs policy field.

  I hope you will find the changes I have made satisfactory.

Reviewer 2 Report

Dear Author: thank you for the opportunity to read your work. The paper addresses an issue is highly relevant and of great interest to the broader public. The findings of the research presented in the paper might be of great interest to researchers and opinion-makers. However, for these findings to be useful and usable, several features of the paper have to be improved, including:

  1. outline of the paper's structure: rather than explaining that the paper is a version of a thesis, the author should focus on more carefully highlighting the research puzzle, the research question and the research outline. Only in this way the reader will know where the paper comes from and where it is headed.  Hence, remove the part on thesis etc. and rethink the introduction.
  2.  The conceptual framework and the research methods used: the paper very clearly uses a specific conceptual framework and specific research methods. A brief, and insufficient, discussion on these two issues has been squeezed in the introduction. And yet, it would be much more convenient to have a specific section on the conceptual framework (a visualization would also help), and on the research method. 
  3. As for the conceptual framework: I miss a clearer conversation justifying the validity of applying Jessop's work in the content of the problem explored in the paper. 
  4. As for the very content of the paper: (a) I miss a clear argument that would explain why the EU member-states (MS) have the leverage to instrumentalize (im)migration and, by default, also national approaches (not really policies) to migration, and (b) thereby expose their 'state power'. I think it is really important. (c) Also, I am not sure which (if any) of the EU MS have developed comprehensive immigration policies... Why not? It is another piece of the puzzle. Finally, (d) I miss a closer consideration of international law and the ways it resonates in the EU MS approaches to (im)migration, i.e. consider the complex legal framework that regulates issues pertaining to migration, including: human rights law, labor law, refugee law, consular law and maritime law. 
  5. Literature: the body of literature used to substantiate the points raised in the paper should be enriched. 

Author Response

Dear Reviewer: 

Thank you very much for reading my article, for your kind words and for your very detailed and helpful comments and suggestions, almost all of which I have tried to attend to. Below, you will find my responses to the points you have raised.

  1. I have duly removed most references to my PhD and I have rewritten the introduction and conclusion to outline the background, content, research problem, empirical materials and research strategy adopted to make sense of developments over time by drawing on official policy documents and legislative acts. I hope that both the research problem and findings now shine through more clearly.
  2. 2 & 3. I have expanded and tried to improve the discussion of the conceptual framework and research methods adopted for this study and to specifically justify using Jessop’s theory on the state and its power(s) to analyse EU policy (sections 2-5) while acknowledging the existence of three different levels: national, EU and intergovernmental (through the Council and bilateral agreements or deals). I believe that I was fortunate to find a theoretical framework that was so well suited to the analysis on the outcomes of institutional workings that I wished to undertake. It seemed tailor-made for this kind of inquiry into the EU as a state-building project and I hope that I have now managed to make that clearer.
4. a) Member states do and have done so by steering policy development within the Council through their representations, first by taking advantage of the intergovernmental bias that applied to the Justice and Home Affairs policy field, and then by including clauses to circumvent limits to state power(s) and enhanced EU scrutiny in legislative instruments. This has made it easier to agree on repressive measures by introducing and expanding measures and procedures for exclusion and surveillance than on safeguarding people, their lives and rights. Moreover, the process has been conducted by giving a technical and operational primacy to interior ministries and law enforcement agencies in steering developments for the benefit of their purposes, treated acritically in an outlook for which “more security” and “more state power” to enforce its aims was (and is) deemed inherently beneficial. Constant pressure in this direction affected the Commission’s outlook. As of 2014/2015, it began to instruct member states to water down legal safeguards for migrants and refugees in an effort to perfect a policy model that has proved dysfunctional for two decades, promoting an authoritarian drift starting from so-called “frontline states”. This paper seeks to illuminate the process through which these conceptual shifts with material impact (also in the form of structures to advance this policy field’s objectives) have taken place. The interplay between levels is crucial, but they are geared to produce similar outcomes in terms of coercive state power(s). I have added discussion of the aspects you raise, which I hope will help. In a different direction, right wing ministers in member states who refuse to comply with Commission instructions by acting even worse than advised or overtly violating international law in broad daylight (Salvini comes to mind, but he is far from alone) are also instrumentalizing immigration policy. b+c) The relationship between migration policy, state power and a drive from different levels (national, EU and intergovernmental) that differs in approach but collectively amounts to a power grab with substantive effects that stray from the formal justifications evoked to develop this policy field are a key interest. The “state project” concept used by Jessop explains the perceived need to create and develop EU security and justice and home affairs structures over time. Governmental approaches to fight irregularised migration at the national level also offer evidence of a trajectory aiming to expand and intensify state power, raising concern about authoritarianism, discrimination, rights and the rule of law. Rather than saying that member states are driving this process, I suggest that it is a two-pronged (or three-pronged if one considers government representations acting collectively -almost like a cartel- through the Council) process involving member states and EU institutions. In pursuit of a policy goal to criminalise mobility from poor and troubled countries, thus helping to entrench and intensify global inequality, the Commission has increasingly shed its role as the guardian of EU treaties. This has been obvious since the end of the Italian Mare Nostrum search-and-rescue operation in the Mediterranean, the unveiling of the European Agenda on Migration and the launch of the hotspot approach in “frontline states” in 2014/2015. It is important for this to be clear: in 2015, pressure that the EU and member states have exercised on third states (like Libya and Morocco) since the late 1990s to commit human rights violations and adopt illegal practices on a large scale against migrants to prevent mobility towards the EU, began being exercised on frontline member states (multiscalar metagovernance, in the article). This point is very important and laden with consequence, so I thank you for making me aware that it was not clear. I have now sought to make this point shine through better. It is unacceptable for third states to be pressured to degrade themselves them based on a policy of which the EU and member states are the sole (supposed) beneficiaries and has purely internal justifications. However, a drive to undermine positive values within the EU in pursuit of migration management is another dimension pointing to this policy field’s damaging substantive impact on states, territories, normative frameworks, people and societies. For comprehensive in relation to national migration policies, I intend well developed or that they have some history and are not something new (like in the UK, France or Belgium). I think states have lost the interest to develop well rounded and comprehensive migration policy frameworks because they fear it might limit what they are allowed to do if there are well established procedures. It is similar to why the five-yearly plans for the AFSJ’s development were no longer produced after the Stockholm Programme. d) Regarding deeper analysis of points pertaining to international law, human rights and the law of the sea, among others, I already had problems of space to convey my findings and would have liked to include more. This article was concerned with a methodological and theoretical approach that I believe has great potential, because it brings together elements that are conveniently decoupled in institutional workings (like the relationship between migration policy goals and the ways in which they are achieved in particular contexts by promoting human rights violations) and insights into motivations and self-interest beyond what is declared. Sadly, this was not the article for that level of detail, but I have covered more legal aspects in the media, academic articles, analyses, reports etc.. 5. I have added several references in support of my claims from different fields (drawn from law, philosophy, migration studies, EU studies, state theory, policy, politics and international relations), some of which were omitted for reasons of length, and I would like to add many more. A wealth of recent academic literature critical of substantive aspects of EU migration policy shows the importance of inquiry into JHA aspects of migration policy and their systemic effects on the EU, member states and third countries. Such effects have evolved from an initial power grab to overcome the limits to what states are allowed to do to people, into a wrecking ball that threatens to undo positive values and principles that the EU proclaimed to uphold.

Thank you very much for your very helpful comments. I hope you will find the changes I have made satisfactory.

Reviewer 3 Report

This is an interesting article looking at the evolution of EU migration policy.  I appreciate the develop of a theoretical framework and applying that framework to primary documents in an effort to explain the evolution of EU policy.  I think the research design is appropriate to the question being asked, with the key issues justified through use of previous literature.  

I had two minor comments on the paper.  First, I would suggest you remove all references to your PhD Thesis in the text of the paper.  I'm not sure it matters to the academic community that this article presents findings from your PhD Thesis.  Your research will stand on its own and is well executed in the research.  I don't want to discourage you from being proud of your Thesis (which I'm sure if outstanding given the quality of this article), but I want you to move this from your Thesis to a journal article.  

My second comment deals with the last paragraph of your introduction.  I'm wondering if it is better suited in section 3 of your paper (making sense of the European migration policy fix).  To me, the inclusion of the discussion of specific migration policy and the inclusion of the table with each of the actions of the Council might be a bit too specific for an introduction.  I think the discussion in lines 83-92 is appropriate for the introduction.  From there on, this seems better suited for section 3 of the paper.  I would also suggest you move the chart of Council actions to section 3 of the paper. 

Overall, I really enjoyed your paper.  I think it adds an interesting dimension to the intersection between politics and migration policy.    

Author Response

Dear Reviewer:   Thank you very much for taking the time to read and comment on the article, and for your kind words.  I have followed your recommendations by moving the specified lines into the new introduction and moving the rest of the paragraph in the suggested section. I have also moved the table of empirical documents further down, to the end of section 3 (l. 328-333), just before the article becomes more reliant on evidence from the documentation. I have removed all mentions of my PhD, apart from a couple of places where it is referenced as a source. Apart from this, the article has undergone extensive revision in response to many comments and suggestions received from other reviewers. I hope you don’t feel I have spoilt it, but I do think some points come across a bit more clearly and it is less unorthodox.  Thank you very much for your encouraging and helpful comments. I hope you will find the changes I have made satisfactory.

Round 2

Reviewer 1 Report

The paper can be published.

Author Response

Thank you very much for your comments and advice.

Reviewer 2 Report

Dear Author:

thank you very much for carefully considering and addressing my comments. The paper has thus been substantially improved and I will only have some minor suggestions, including:

  1. In the abstract, and then also in the body text, you use the term "European migration policies". I suggest that you be more precise, i.e. European Union member-states migration policies or something of this kind.
  2. As for the literature I would add a few items just to stress the relevance of the topic, and substantiate the notion of contingencies specific to the formulation of approaches (because frequently incoherent sets of rules and regulations) to migration etc. Please, consider these items for instance: 
  3. https://onlinelibrary.wiley.com/doi/10.1111/imig.12648
  4. https://www.tandfonline.com/doi/full/10.1080/1369183X.2017.1341718
  5. https://www.tandfonline.com/doi/full/10.1080/13501763.2017.1325920

Author Response

Dear reviewer:

Thank you so much for your detailed comments and advice, as well as the valuable literature towards which you directed my attention. I have taken your advice and changed references to European migration policies to include the EU and member states in several places where this was appropriate (lines 4, 433, 472, 786). I have refrained from doing so, when it was a section that specifically focused the EU side of this process. I have also read the articles that you suggested, which enriched my understanding and developed some themes that I hinted at, so I incorporated one or a few relevant thoughts from each of them that I felt would contribute to better place this article within existing academic literature (l. 209-214, 538-547, 600-608). I also proofread the whole article again, making very slight corrections, mainly for grammatical coherence or typos, added a couple of bibliographic references (and those for the articles you suggested) that were missing, removed one and added another acronym, and included brief acknowledgements at the end.